# Eigenvalues of the Hessian in Deep Learning: Singularity and Beyond

**Levent Sagun**
Mathematics Department
New York University
`sagun@cims.nyu.edu`

**Léon Bottou**
Facebook AI Research
New York
`leon@bottou.org`

**Yann LeCun**
Computer Science Department
New York University
`yann@cs.nyu.edu`

## Abstract

We look at the eigenvalues of the Hessian of a loss function before and after training. The eigenvalue distribution is seen to be composed of two parts, the bulk which is concentrated around zero, and the edges which are scattered away from zero. We present empirical evidence for the bulk indicating how over-parametrized the system is, and for the edges that depend on the input data.

## 1 Introduction

Given a (piece-wise) differentiable loss function, and a gradient based algorithm to minimize it, the knowledge of the second order information about it can tell us quite a bit about how the landscape looks like, and how we could modify our algorithm to make it go faster and find better solutions. But, one of the biggest challenges in second order optimization methods is in accessing that second order information itself. In particular, in deep learning there have been many proposals to accelerate training using second order information. Ngiam et al. (2011) has an in depth review of some of the proposals for approximating the Hessian of the loss function. Nevertheless, given the computational complexity of the problems at hand, it is hard to acquire information on what the actual Hessian looks like. This work is part of a series of papers that explore the data-model-algorithm connection along with Sagun et al. (2014; 2015) and it builds on top of the intuition developed in LeCun et al. (2012). We also note that the singularity of the Fisher information matrix have been explored (see for instance Watanabe (2007)). In another recent work, Dauphin et al. (2014) investigate saddle points of the landscape, in particular, they locate saddle points that are near the training path. In this work, however, we strictly focus on the Hessian of the loss function at the exact point of the training. We train the main examples using gradient descent. We perform our calculations of the Hessian using the implementation for the exact Hessian vector product that has been introduced in Pearlmutter (1994). And we find two new observations: one where the eigenvalues are zero, and one where we have a large, positive, and discrete set of eigenvalues.

In this short note, we show how the data and the architecture depends on the eigenvalues of the Hessian of the loss function. In particular, we observe that the top discrete eigenvalues depend on the data, and the bulk of the eigenvalues depend on the architecture. Furthermore, as we keep growing the size of the network, we observe that the discrete part that depends on data remains the same, but the concentration around zero sharpens.

There are various conclusions and implications of this singularity. Recent research suggest new insights into convergence properties of gradient based algorithms in non-convex systems (Lee et al., 2016; Hardt et al., 2015). The results come together with their implications on neural networks. However, the proofs require the system at hand to be non-degenerate. An immediate conclusion of our observation is that the Hessian of the loss function is very singular. Therefore, a lot of the theory and methodology that assumes non-singular Hessian cannot be applied without an appropriate modification.

## 2 THE CASE OF THE FULLY-CONNECTED NETWORK

### 2.1 MNIST WITH INCREASING SIZES OF HIDDEN LAYERS

We calculate the exact Hessian of the loss function of a network with two hidden layers. The inputs are 1000 randomly selected examples of $28 * 28$ MNIST data, the network has one hidden layer with ReLU nonlinearity, the top layer has a softmax and a negative log likelihood loss function at the end. We train the system with gradient descent (i.e. with minibatch size equal to the number of examples). We plot the histogram of the eigenvalues of the Hessian for a varying number of hidden units after convergence. The Hessian at the end of the training turns out to be extremely singular, and increasing the number of units in hidden layers only add to the singularity of the Hessian (see figure 1). The effect of the training on the eigenvalue spectrum of the model with 10 hidden-units is visible when comparing figure 1 and figure 2 (left).

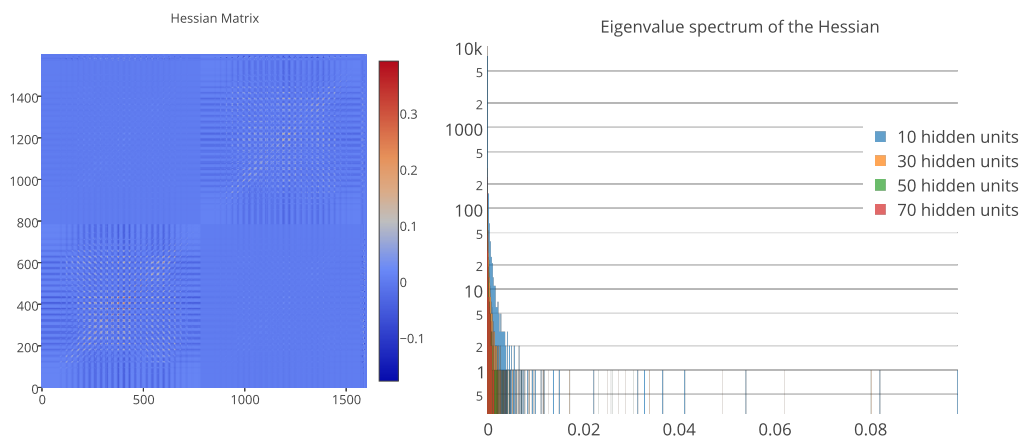

Figure 1: (left) Full Hessian matrix for a 784-2-10 system after convergence. (right) Eigenvalue profile for increasingly bigger networks. For $k$ hidden networks there are $(784 + 1) * k + (k + 1) * k + (k + 1) * 10$ eigenvalues.

### 2.2 VARYING THE DATA

To demonstrate how the eigenvalue distribution may depend on data itself, we keep the same architecture and change the inputs to random patterns. Initially, a random point in the weight space is selected, and we calculated the Hessian without any training (first two histograms of figure 2). After training the system until the norm of the gradient is close to zero. We again calculate the exact Hessian and plot the histogram of their eigenvalues which can be seen in the last one in figure 2.

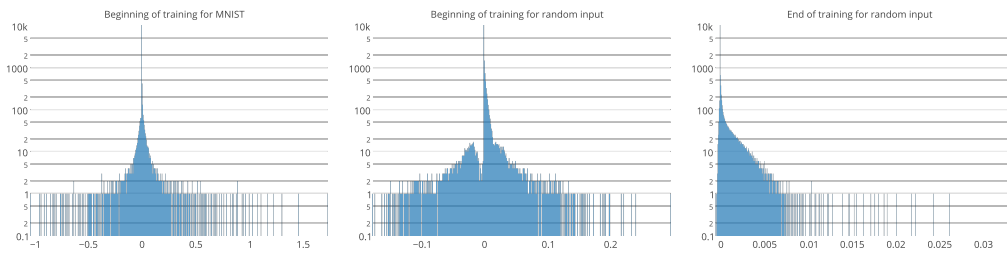

Figure 2: Comparing random input (last two) with the MNIST data (first). Initial eigenvalue profiles are very different, as well as the final profile when compared to figure 1.

## 3    A SIMPLER CASE

In this section, we will repeat the same experiment in two-dimensional data, in an attempt to understand better the connection between the data and the spectrum of the Hessian. A simple figure can be seen in figure 3. We create two Gaussian blobs, centered at $(1,1)$ and $(-1,-1)$, and first we keep the standard deviation the same, and increase the network size.

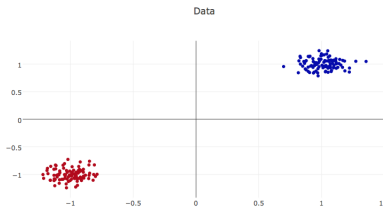

Figure 3: The input data for the simple case.

The network architecture is similar, this time with two hidden layers and a fully connected network with ReLU nonlinearities including a softmax at the top layer combined with a negative log-likelihood loss function. We train the system with gradient descent with constant step size. At the end of the training the norm of the gradient is at the order of $10^{-4}$.

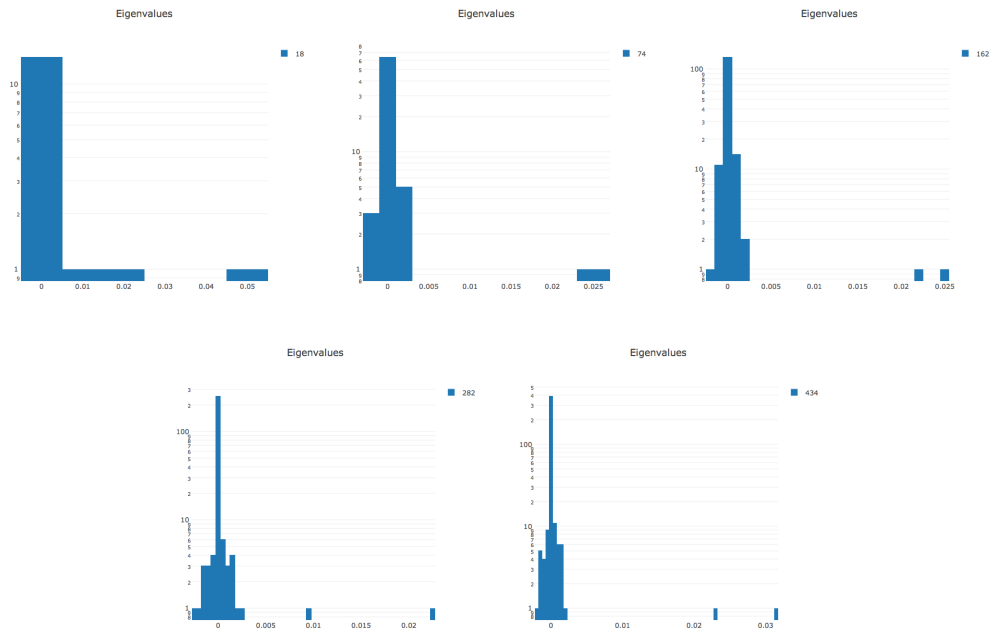

Figure 4: Increasing the network size: Systems with 18, 74, 162, 282, and 434 parameters, respectively. And a network with MSE loss.

There are two eigenvalues that are isolated, and away from the bulk of the spectrum. Increasing the network only adds to the concentration of eigenvalues at and around zero (see figure 4). To give an insight into how the Hessian's themselves look like, in figure 6 we plot the full Hessian matrices for three of the systems above after training.

Moreover, this property of the singular and discrete parts is not specific to the log loss. In figure 5, we plot the histogram for a system that is trained on the same data as in figure 3, and the same

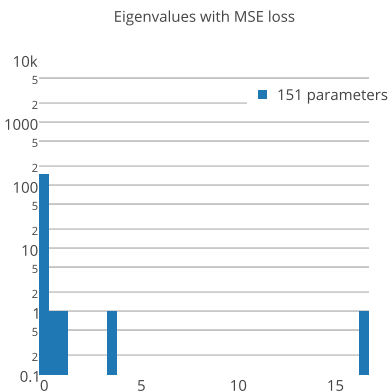

Figure 5: Spectrum for the loss with the mean square loss.

architecture. But the training is carried out with the mean square loss rather than the negative log-likelihood. Consistent with our previous observations, we still see the same discrete, data-dependent part, and the part that is at zero.

## Rotated Hessian Matrix

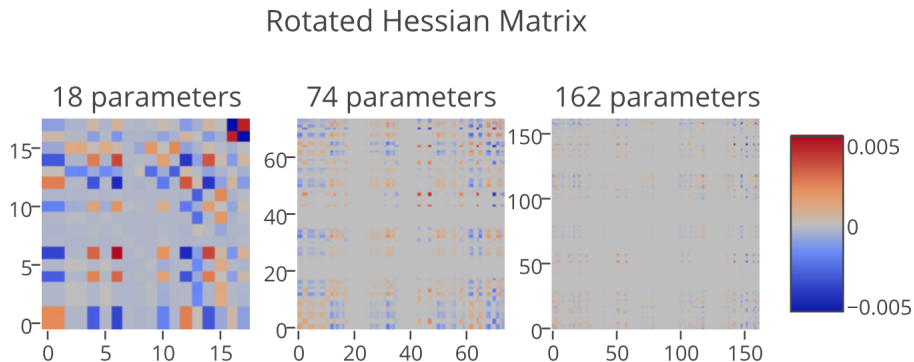

Figure 6: Hessian heatmaps for 18, 74 and 162 paramters systems after training. The plots are 90 degrees rotates counter-clock wise.

The training procedure itself acts like a process by which the eigenvalues concentrate at zero. To demonstrate this in further detail we calculated the full Hessian peridoically throughout the training. In figure 7 we plot the eigenvalue profile as the training progresses.

All of the training has been done with random initializations on the weight space with the same standard deviation. In other words, initial points are randomly chosen on the surface of a sphere with a fixed radius given the total number of parameters. This begs the question of the effect of the choice of the initial point. Therefore, now we fix the network size, and repeat the experiment with different random initializations over 5K times. In figure 8 we plot the fluctuations of the top eigenvalue.

The next question is how the spectrum responds to the increased complexity of data. The notion of complexity for a given dataset can be tricky to describe, here we use a loose notion of complexity to point out the fact that the more complex data is the less separable one. To this end, we keep the architecture the same, and increase the standard deviation of the two Gaussian blobs. They are still centered at the same two points, but it becomes harder to separate them as they merge together. Gradient descent still converges to a low-cost value, but the error is higher, and it can't learn how to separate them perfectly as the blobs merge together. In figure 9, we observe that the top two

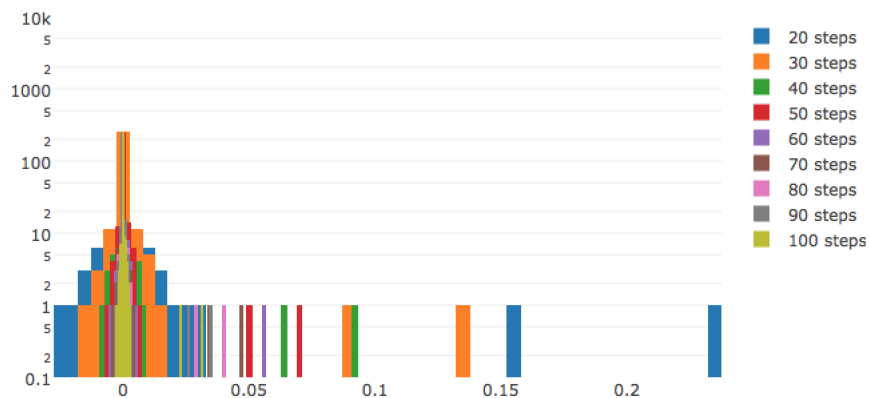

Figure 7: Eigenvalue profile during the training procedure.

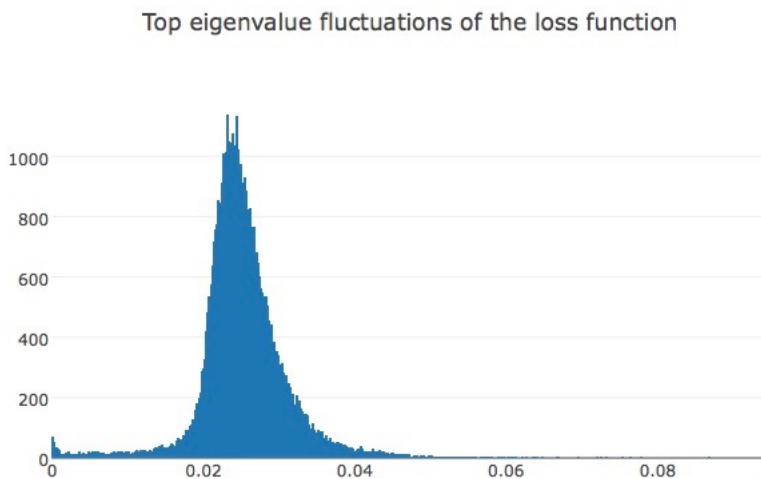

Figure 8: Top eigenvalue fluctuations over 5000 runs of the same system with same data and algorithm but different initial points.

eigenvalues grow significantly, and beyond its natural fluctuations due to the initialization. We also note that the norm of the weights are similar for all the cases, therefore the growth in the sizes of eigenvalues can not solely be accounted for the growth in weights.

## 4 CONCLUSION

We show that the Hessian of the loss functions in deep learning is degenerate. This has implications on the theoretical work which requires improvements in its premises. One such step has been taken in Panageas & Piliouras (2016) in relaxing the isolated singularity condition that was assumed in Lee et al. (2016). From a practical point of view this has multiple implications:

- The landscape may be flat beyond the notion of wide basins.
- Training stops at a point that has a small gradient. The norm of the gradient is not zero, therefore it does not, technically speaking, converge to a critical point.
- There are still negative eigenvalues even when they are small in magnitude.

This suggests that we may be able to look beyond the classical notions of basins when exploring the energy landscapes of loss functions. Next obvious question is to find low energy paths between

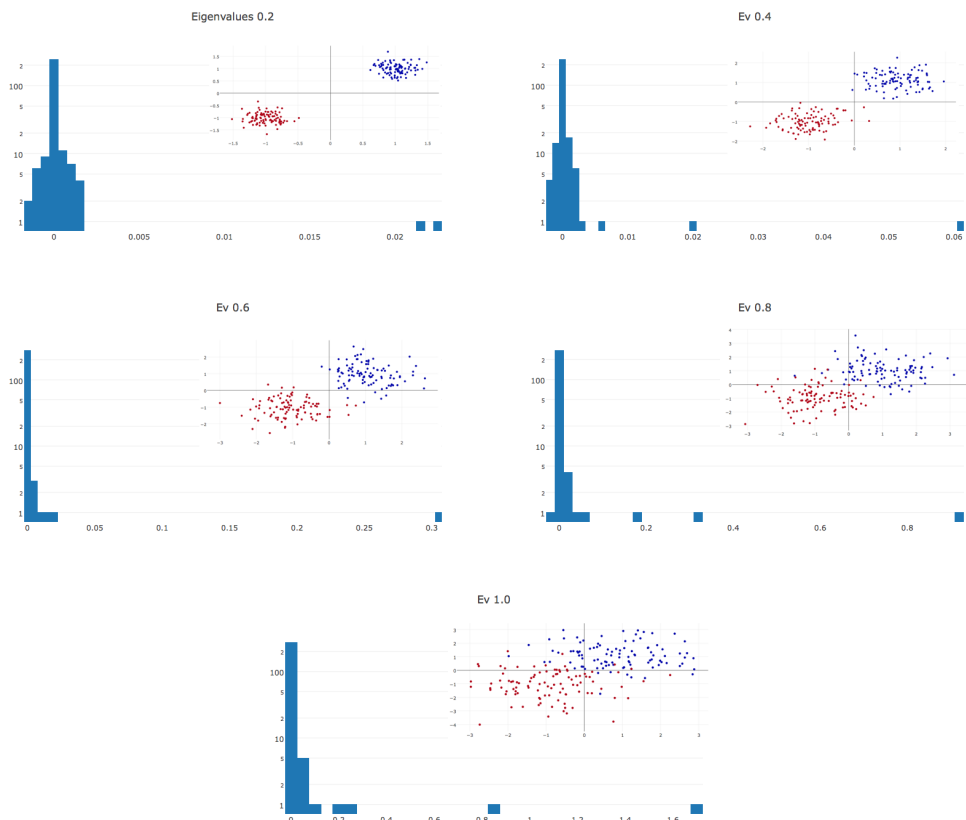

Figure 9: Response of the top eigenvalues to the increasingly less-separable data. The numbers on top of the figures indicate the standard deviation of the Gaussian blobs. Their means are kept the same at $(1, 1)$ and $(-1, -1)$, respectively.

solutions to show the kind of flatness in such landscapes. This will be explored in a subsequent work in the same series.

We also demonstrate the two phases of the spectrum, one that is concentrated around zero that depends on the size of the model, and the second part that is away from the bulk of the spectrum, that is isolated and depends on the data.

This kind of two-phased non-degeneracy can, in fact, be a desirable property. A degenerate Hessian implies locally flat regions. A degenerate Hessian at the scale that we observe in deep learning may imply flat regions across space, at the global scale.

- We can devise separate methods for the directions that correspond to the top eigenvalues.
- We can take advantage of the directions that correspond to the zero or small eigenvalues by attempting to find paths of low energies in the weight space.

As a first step to the last item, initial experiments are promising: Let's take a random point on the weight space and train two systems from that point: (1) with gradient descent, and (2) with stochastic gradient descent. At each step, take a straight line between the two points and interpolate the cost value. The resulting profile is completely flat even when the points keep diverging from one another. Next, take two random initial points on the weight space, so now they are orthogonal to each other. And train two systems with different shuffling of data for SGD. This time one would expect the line interpolation to give arbitrary values since the initial points are completely orthogonal, surprisingly, the line interpolation also decreases albeit not as flat as the previous one. Further considerations on

connecting paths between solutions in the weight space of loss functions can be found in Freeman & Bruna (2016).

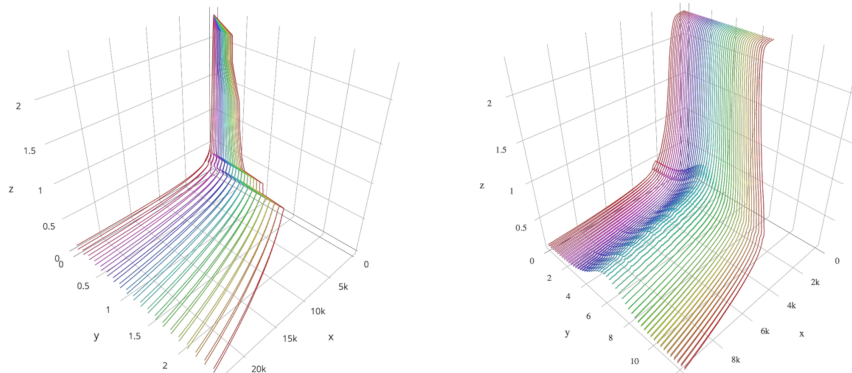

Figure 10: $z$-axis is the distance between points. The left most and right most curves in each plot are actual training profiles, and the lines in between are interpolations only. (left figure) same initial point (right figure) random (hence orthogonal) initial points.

ACKNOWLEDGMENTS

We would like to thank Afonso Bandeira, Yann Dauphin, Ruoyu Sun, Arthur Szlam and Soumith Chintala for valuable discussions. We also thank the reviewers for valuable feedback. Part of the research has been conducted when the first author was an intern at FAIR.

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
