# Peer review of "Eigenvalues of the Hessian in Deep Learning: Singularity and Beyond"

_ICLR 2017 — rejected_

[Official Review · AnonReviewer2 · rating 4 · confidence 5 · 13 Dec 2016]
**This paper presents empirical evidence for the singularity of the Hessian. This works has interesting experiments and observations but the paper needs more work and it is not a complete work.**

Studying the Hessian in deep learning, the experiments in this paper suggest that the eigenvalue distribution is concentrated around zero and the non zero eigenvalues are related to the complexity of the input data. I find most of the discussions and experiments to be interesting and insightful. However, the current paper could be significantly improved.

Quality:
It seems that the arguments in the paper could be enhanced by more effort and more comprehensive experiments. Performing some of the experiments discussed in the conclusion could certainly help a lot. Some other suggestions:
1- It would be very helpful to add other plots showing the distribution of eigenvalues for some other machine learning method for the purpose of comparison to deep learning.
2- There are some issues about the scaling of the weights and it make sense to normalize the weights each time before calculating the Hessian otherwise the result might be misleading.
3- It might worth trying to find a quantity that measures the singularity of Hessian because it is difficult to visually conclude something from the plots.
4- Adding some plots for the Hessian during the optimization is definitely needed because we mostly care about the Hessian during the optimization not after the convergence.

Clarity:
1- There is no reference to figures in the main text which makes it confusing for the reading to know the context for each figure. For example, when looking at Figure 1, it is not clear that the Hessian is calculated at the beginning of optimization or after convergence.
2- The texts in the figures are very small and hard to read.

[Public Comment · (anonymous) · rating 4 · confidence 5 · 16 Dec 2016 (modified: 17 Dec 2016)]
**Interesting empirical observations, but unfortunately no theory**

The work presents some empirical observations to support the statement that “the Hessian of the loss functions in deep learning is degenerate”. But what does this statement refer to? To my understanding, there are at least three interpretations:

(i) The Hessian of the loss functions in deep learning is degenerate at any point in the parameter space, i.e., any network weight matrices.

(ii) The Hessian of the loss functions in deep learning is degenerate at any critical point.

(iii) The Hessian of the loss functions in deep learning is degenerate at any local minimum, or any global minimum.

None of these interpretations is solidly supported by the observations provided in the paper.

More comments are as follows:

1) The authors state that “we don’t have much information on what the actual Hessian looks like.” Then I just wonder what Hessian is investigated. Is it the actual one or approximate one? Please clarify and provide the references for computing the actual Hessian.

2) It is not clear whether the optimization was done by a batch gradient descent algorithm, i.e., batch back propagation (BP) algorithm, or a stochastic BP algorithm. If the training was done via a stochastic BP algorithm, it is hard to conclude that the the Neural Network has been trained to its local minimum. When it was done by a full-batch BP algorithm, what was the accumulating point? Was it local minimum or global minimum?

3) Since the negative log likelihood function was used as at the end of training, it is essentially a joint learning approach in both the Newton weight matrices and the negative log likelihood vector. Certainly, the whole loss function is not convex in these two parameters. But if least squares error function is used at the end, would it make any difference in claiming the degeneracy of the Hessian?

4) Finally, the statement “There are still negative eigenvalues even when they are small in magnitude” is very puzzling. Potential reasons are:
(a) If the training algorithm did converge, the accumulating points were not local minima, i.e., they were saddle points.
(b) Training algorithms did not converge, or have not converged yet.
(c) The calculation of the actual Hessian might be inaccurate.

[Official Review · AnonReviewer3 · rating 4 · confidence 5 · 19 Dec 2016]
**interesting insights, but lack of control experiments**

The paper analyzes the properties of the Hessian of the training objective for various neural networks and data distributions. The authors study in particular, the eigenspectrum of the Hessian, which relates to the difficulty and the local convexity of the optimization problem.

While there are several interesting insights discussed in this paper such as the local flatness of the objective function, as well as the study of the relation between data distribution and Hessian, a somewhat lacking aspect of the paper is that most described effects are presented as general, while tested only in a specific setting, without control experiments, or mathematical analysis.

For example, regarding the concentration of eigenvalues to zero in Figure 6, it is unclear whether the concentration effect is really caused by training (e.g. increasing insensitivity to local perturbations), or the consequence of a specific choice of scale for the initial parameters.

In Figure 8, the complexity of the data is not defined. It is not clear whether two fully overlapping distributions (the Hessian would then become zero?) is considered as complex or simple data.

Some of the plots legends (Fig. 1 and 2) and labels are unreadable in printed format. Plots of Figure 3 don't have the same range for the x-axis. The image of Hessian matrix of Figure 1 does not render properly in printed format.

[Official Review · AnonReviewer1 · rating 3 · confidence 4 · 20 Dec 2016]
**Important problem, but should be better situated in related work**

This paper investigates the hessian of small deep networks near the end of training. The main result is that many eigenvalues are approximately zero, such that the Hessian is highly singular, which means that a wide amount of theory does not apply.

The overall point that deep learning algorithms are singular, and that this undercuts many theoretical results, is important but it has already been made: Watanabe. “Almost All Learning Machines are Singular”, FOCI 2007. This is one paper in a growing body of work investigating this phenomenon. In general, the references for this paper could be fleshed out much further—a variety of prior work has examined the Hessian in deep learning, e.g., Dauphin et al. “Identifying and attacking the saddle point problem in high dimensional non-convex optimization” NIPS 2014 or the work of Amari and others.

Experimentally, it is hard to tell how results from the small sized networks considered here might translate to much larger networks. It seems likely that the behavior for much larger networks would be different. A reason for optimism, though, is the fact that a clear bulk/outlier behavior emerges even in these networks. Characterizing this behavior for simple systems is valuable. Overall, the results feel preliminary but likely to be of interest when further fleshed out.

This paper is attacking an important problem, but should do a better job situating itself in the related literature and undertaking experiments of sufficient size to reveal large-scale behavior relevant to practice.

[Author Response · Levent Sagun · 20 Jan 2017]
**Revisions**

In light of the constructive feedback, we revised the paper with multiple edits and figure revisions. We also included a new experiment showing that the phenomena that we observe is not specific to the log-loss, that it holds in the case of MSE loss, as well. 

We also noticed that the title can be misleading in that it may suggest that our work's focus is on the singularity of Hessian only. Indeed, the good part of the work is related to the singularity, however, we have a second main message regarding the eigenvalue spectrum: the discrete part depends on data. We revised the title and parts of the body of the text to emphasize this point.

[Final Decision · Program Chairs · 06 Feb 2017]
**ICLR committee final decision**

This is quite an important topic to understand, and I think the spectrum of the Hessian in deep learning deserves more attention. However, all 3 official reviewers (and the public reviewer) comment that the paper needs more work. In particular, there are some concerns that the experiments are too preliminary/controlled and about whether the algorithm has actually converged. One reviewer also comments that the work is lacking a key insight/conclusion. I like the topic of the paper and would encourage the authors to pursue it more deeply, but at this time all reviewers have recommended rejection.